

# Fibroblast growth factors as tissue repair and regeneration therapeutics

Quentin M. Nunes[1],*, Yong Li[2],*, Changye Sun[2],*, Tarja K. Kinnunen[3] and David G. Fernig[2]

[1] Department of Molecular and Clinical Cancer Medicine, NIHR Liverpool Pancreas Biomedical Research Unit, University of Liverpool, Liverpool, United Kingdom

[2] Department of Biochemistry, Institute of Integrative Biology, University of Liverpool, Liverpool, United Kingdom

[3] Department of Biology, School of Applied Sciences, University of Huddersfield, Huddersfield, United Kingdom

* These authors contributed equally to this work.

Corresponding author
David G. Fernig, dgfernig@liv.ac.uk

## ABSTRACT

Cell communication is central to the integration of cell function required for the development and homeostasis of multicellular animals. Proteins are an important currency of cell communication, acting locally (auto-, juxta-, or paracrine) or systemically (endocrine). The fibroblast growth factor (FGF) family contributes to the regulation of virtually all aspects of development and organogenesis, and after birth to tissue maintenance, as well as particular aspects of organism physiology. In the West, oncology has been the focus of translation of FGF research, whereas in China and to an extent Japan a major focus has been to use FGFs in repair and regeneration settings. These differences have their roots in research history and aims. The Chinese drive into biotechnology and the delivery of engineered clinical grade FGFs by a major Chinese research group were important enablers in this respect. The Chinese language clinical literature is not widely accessible. To put this into context, we provide the essential molecular and functional background to the FGF communication system covering FGF ligands, the heparan sulfate and Klotho co-receptors and FGF receptor (FGFR) tyrosine kinases. We then summarise a selection of clinical reports that demonstrate the efficacy of engineered recombinant FGF ligands in treating a wide range of conditions that require tissue repair/regeneration. Alongside, the functional reasons why application of exogenous FGF ligands does not lead to cancers are described. Together, this highlights that the FGF ligands represent a major opportunity for clinical translation that has been largely overlooked in the West.

## OVERVIEW

In unicellular organisms the unit of natural selection is the cell, whereas in multicellular animals natural selection operates on the organism. This is a very profound difference. The driver is likely to have been simple: there is a limit on the complexity of an individual cell, beyond which it is no longer robust. However, greater organism complexity allows
new ecological niches and lifestyles to be exploited. Natural selection has given rise to multicellularity and cell specialisation, as a means to allow a high level of organism complexity in concert with simple and robust cells. This requires a deep functional integration of cells in the organism, achieved through cell communication, which occurs by cells delivering information through the synthesis and secretion of signalling molecules into the extracellular space; these then elicit signals in cells possessing appropriate receptor systems. The entire biochemical landscape, from ions and small molecules to proteins and polysaccharides is used to generate the repertoire of signalling molecules.

In multicellular animals, proteins are common currency in cell communication and are used to transmit information between cells in the organism both locally (intra-, auto-, juxta- and paracrine) and systemically (endocrine). Local transmission of information may be mediated by a soluble, secreted protein, or by a protein anchored in the extracellular matrix or on the plasma membrane of a neighbouring cell. The exploitation of proteins for cell communication by multicellular animals provides access to a very subtle language. This subtlety arises in part from the fact that an individual protein species may have many different isoforms (from splice variants to glycoforms), localisations and interacting partners. Each subset of molecular interactions that an individual protein species can partake in may elicit completely different, sometimes opposing, cellular responses, and, moreover, may change the distance over which communication occurs, e.g., paracrine *versus* endocrine.

Most therapeutics can be considered to manipulate cell communication, with varying aims, such as reducing overactive communication channels in cancer and inflammatory diseases, or increasing particular channels of communication for tissue repair and regeneration. Clearly, manipulating cell communication therapeutically is not without danger, since the opposite of the desired effect may occur. Less obvious, but an important focus of this review, is that the drivers of scientific discovery can narrow dramatically how a particular communication system is used clinically. This is exemplified by the fibroblast growth factor (FGF) family, which contributes to the regulation of virtually all aspects of development and organogenesis, and after birth in many natural processes of tissue repair and the endocrine regulation of particular facets of organism physiology. In the West, the major focus of clinical translation has been on developing inhibitors of FGF-mediated cell communication for use in cancer therapy. In the East, particularly in China and to an extent in Japan, a major focus has been to use FGFs in regenerative/repair medical settings, differences that have their roots in a combination of research history and research aims. To increase awareness of this work, we summarise a number of published clinical reports to illustrate the breadth and depth of the successful clinical applications of FGFs.

## THE DISCOVERY OF FGF LIGANDS AND THEIR ACTIVITIES

Historically, the growth factor activity was the first to be identified. In hindsight, the paper of *Trowell & Willmer (1939)*, which measured the mitogenic activity of saline extracts of different tissues from the chick can be considered to be the first FGF paper

(*Trowell & Willmer, 1939*)–the activity they isolated from brain would consist of FGF-1, FGF-2 (*Burgess & Maciag, 1989*), as well as other growth factors active on fibroblasts, (e.g., pleiotrophin (*Courty et al., 1991*)), and that from other tissues largely FGF-2 (*Burgess & Maciag, 1989*; *Fernig & Gallagher, 1994*). Over 30 years later a growth factor activity that stimulated the growth of a fibroblast cell line was identified in partially purified extracts from bovine pituitary. It was called "fibroblast growth factor", simply due to the assay used to measure activity (*Rudland, Seifert & Gospodarowicz, 1974*). Though an unsatisfactory name, because FGFs do far more than stimulate fibroblast growth and in a considerable number of instances they do not even possess this activity, the label has stuck (see *Burgess & Maciag, 1989*) for other early names and a brief overview of the discovery of FGF-1 and FGF-2). A great deal of the early work on FGFs, including that of (*Trowell & Willmer, 1939*), was from a cancer perspective, driven by the idea that uncontrolled proliferation is a hallmark of cancers and so growth factors must have a key role to play. Moreover, the ambition to cure cancers provided funding for this and much subsequent work on FGFs and other growth factors. This was not misplaced, since the analysis of experimental tumours and of activities capable of transforming cells *in vitro* enabled the discovery of some, but not all of FGFs -3 to -9 (summarised in *Burgess & Maciag (1989)* & *Fernig & Gallagher (1994)*) and there are a number of successful FGF receptor (FGFR) inhibitors in oncology (*Carter, Fearon & Grose, 2015*; *Turner & Grose, 2010*).

The interaction with heparin was key to the successful purification of FGF-1 and -2 (*Maciag et al., 1984*; *Shing et al., 1984*), and was translated into work on the interaction of these FGFs with the glycosaminoglycan heparan sulfate (HS) in the pericellular and extracellular matrix (e.g., *Vlodavsky et al., 1987*). The FGF receptor (FGFR) tyrosine kinases were then identified and, soon after, the dependence of the growth factor activity of FGFs on heparan sulfate (*Rapraeger, Krufka & Olwin, 1991*; *Yayon et al., 1991*) was discovered. This provided a framework within which to understand function, heparan sulfate controlled the transport of FGFs between cells and was a part of a dual receptor (heparan sulfate + FGFR) signalling system. Subsequently, some FGFs were found to not bind heparan sulfate, but to interact with a protein co-receptor, Klotho; these FGFs do not elicit a growth factor response, but instead are endocrine hormones (*Belov & Mohammadi, 2013*; *Kuro-o et al., 1997*; *Martin, David & Quarles, 2012*). A further set of FGF proteins, the FGF homology factors or FHFs, are wholly intracellular and do not interact with any of the extracellular receptors and partners of FGFs. As such they are not directly part of the FGF cell communication system and lie outside the scope of this review (for review see *Goldfarb, 2005*).

## THE FGF COMMUNICATION SYSTEM: MOLECULES AND STRUCTURE

The core of the FGF communication system comprises a family of ligands, the FGFs, a family of cell surface signal transducing receptors, the FGFRs, and two distinct co-receptors, the Klothos and the glycosaminoglycan heparan sulfate, which is the physiologically relevant polysaccharide; heparin is often used as its experimental proxy, but has important structural differences.

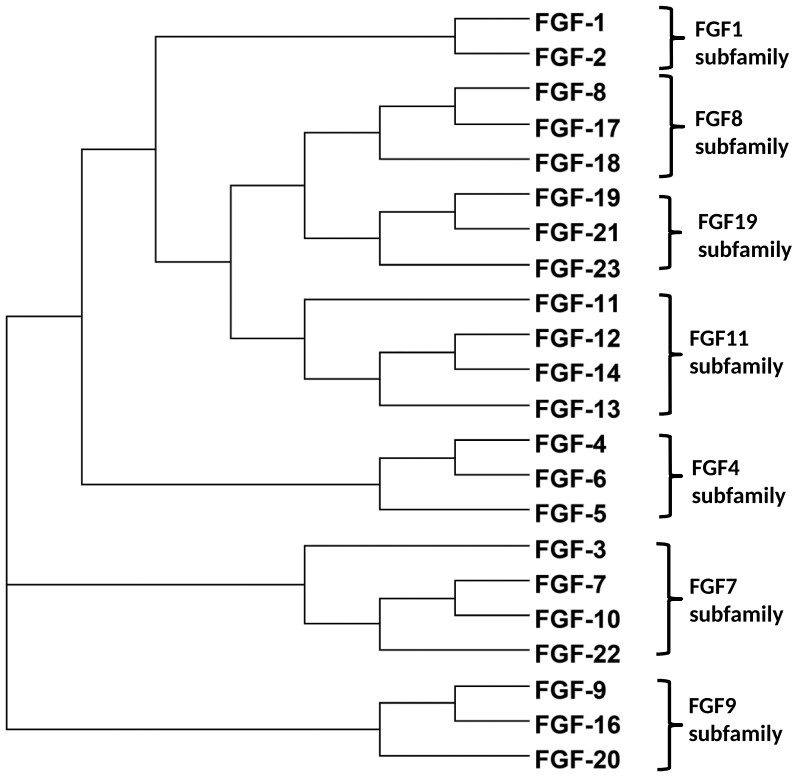

**Figure 1 Phylogenetic relationship of the FGFs based on amino acid sequence.** According to amino acid sequence, dendroscope was used to show that FGF family is divided into seven subfamilies. The branch lengths relates directly to the evolutionary relationship of FGFs.

## The FGF ligand family

Phylogenetic analysis of human protein sequences indicates that there are seven FGF subfamilies: FGF1 and FGF2 (FGF1 subfamily); FGF4, FGF5 and FGF6 (FGF4 subfamily); FGF3, FGF7, FGF10 and FGF22 (FGF7 subfamily); FGF8, FGF17 and FGF18 (FGF8 subfamily); FGF9, FGF16 and FGF20 (FGF9 subfamily); FGF11, FGF12, FGF13 and FGF14 (FGF11 subfamily); FGF19, FGF21, and FGF23 (FGF19 subfamily) (Fig. 1). The members of FGF8, FGF9, FGF11 and FGF19 subfamilies are consistent between the phylogenetic analysis and the gene location analysis. However, FGF5 and FGF3 are indicated to be members of FGF4 and FGF7 subfamilies by the analysis of gene location on chromosomes (Horton et al., 2003; Itoh, 2007; Itoh & Ornitz, 2008; Itoh & Ornitz, 2011). The phylogenetic relationship based on sequence maps to functional similarities of the FGFs (Ornitz et al., 1996; Xu et al., 2013; Zhang et al., 2006) and it is in this context that FGF subfamilies will be discussed here.

## FGF ligand structure

The molecular weight of FGFs range from 17 to 34 kDa in vertebrates, whereas it reaches to 84 kDa in *Drosophila*. All FGFs share an internal core of similarity with 28 highly conserved, and six invariant amino acid residues (Ornitz, 2000). X-ray crystallography of FGFs shows that the FGF family possesses a similar folding pattern to the interleukins

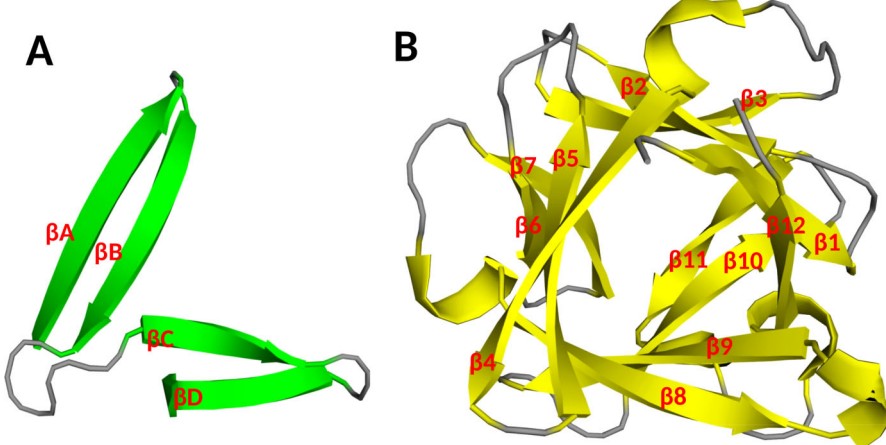

**Figure 2 Schematic diagram of the core structure unit of the beta-trefoil.** PDB ID: 2FGF (*Zhang et al., 1991*). (A) The first ascending strand (βA) is connected to a descending strand (βB). The following "horizontal" strand (βC) finishes by returns strand (βD). (B) Three of these units arranged around a pseudo three-fold axis of symmetry form the β trefoil.

IL-1β and IL-1α (*Zhu et al., 1991*), a β trefoil structure, formed by three sets (Fig. 2A) of four β strands connected by loops (Fig. 2B) (*Zhang et al., 1991*). A variety of studies have demonstrated that the primary heparan sulfate binding site of FGF2 is formed by the strand β1/β2 loop, strands β10/β11 loop, strand β11 and strands β11/β12 loop (Fig. 3B) (*Baird et al., 1988*; *Faham et al., 1996*; *Li et al., 1994*; *Thompson, Pantoliano & Springer, 1994*; *Zhang et al., 1991*). The receptor binding site involves the strands β8-β9 loop and is distinct from the primary heparan sulfate binding site (Figs. 3A and 3B). This indicates that the binding to receptor and to heparan sulfate are physically separated (*Itoh & Ornitz, 2004*; *Ornitz & Itoh, 2001*; *Zhang et al., 1991*). Secondary heparan sulfate binding sites are also present in many FGFs and their position on the surface of the ligands may follow their sequence phylogenetic relationship (*Ori et al., 2009*; *Xu et al., 2012*) (Figs. 3B and 3C).

## Receptors: Heparan sulfate and FGFR

### Heparan sulfate

Proteoglycans are O-glycosylated proteins, such as perlecan, glypicans and syndecans (*Taylor & Gallo, 2006*; *Yung & Chan, 2007*). The heparan sulfate chains bind and regulate the function of over 435 extracellular proteins, including the paracrine FGFs (*Gallagher, 2015*; *Ori, Wilkinson & Fernig, 2008*; *Ori, Wilkinson & Fernig, 2011*; *Xu & Esko, 2014*). The proteoglycan core proteins are synthesized on the rough endoplasmic reticulum and then transported to the Golgi apparatus where the glycosaminoglycan chains are synthesised (*Yanagishita & Hascall, 1992*). The glycosaminoglycan chains are linear polysaccharides mainly consisting of repeating disaccharide units (Fig. 4A) (*Gallagher, 2015*; *Ori, Wilkinson & Fernig, 2008*; *Taylor & Gallo, 2006*; *Xu & Esko, 2014*). The members of the glycosaminoglycan family are heparan sulfate, chondroitin sulfate (CS), dermatan sulfate (DS), hyaluronan (HA) and keratan sulfate (KS) (*Delehedde et al., 2001*).

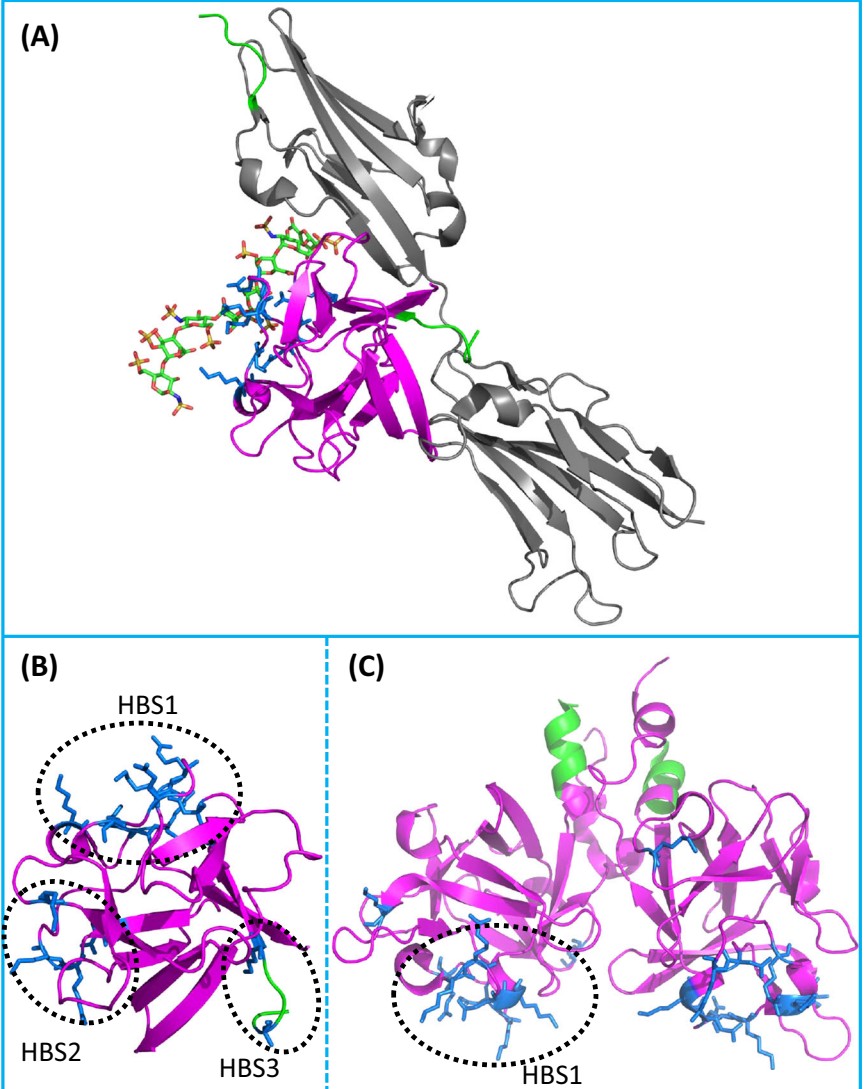

**Figure 3 FGF interactions with FGFR and heparin/heparan sulfate.** (A) Ternary structure of FGF-FGFR-heparin complex (1FQ9 (*Schlessinger et al., 2000*)). FGFs interact with the D2 and D3 domain and the linker between these two domains. A heparin octasaccharide, binds to the conserved canonical binding site on FGFs, which is opposite to the N-terminal, and to the basic canyon in the FGFR. (B) Heparin binding sites of FGF2 (1FQ9) identified by a selective labelling approach (*Ori et al., 2009*). Three binding sites were recognised: the canonical binding site (HBS1), and two secondary and relatively weaker binding sites (HBS2 and HBS3). (C) Heparin binding site of FGF9 (1G82 (*Hecht et al., 2001*)). Only the conserved HBS1 was identified, indicating that FGF9 does not possess secondary polysaccharide binding sites (*Xu et al., 2012*), subsequently confirmed in biophysical experiments (*Migliorini et al., 2015*). Green indicates the N-terminal of the proteins. Grey is FGFR1. Magenta are FGFs (FGF2 in B and FGF9 in C). The residues in blue are the heparin binding sites of the FGFs.

Heparan sulfate is made of repeating disaccharide units of glucuronic acid linked to N-acetylglucosamine (Fig. 4A). In the Golgi apparatus, the synthesis of heparan sulfate chains is started by the assembly of a tetrasaccharide linkage onto a serine residue of the core protein by four enzymes acting sequentially (Xyl transferase, Gal transferase I and II and GalA transferase); the repeat disaccharide

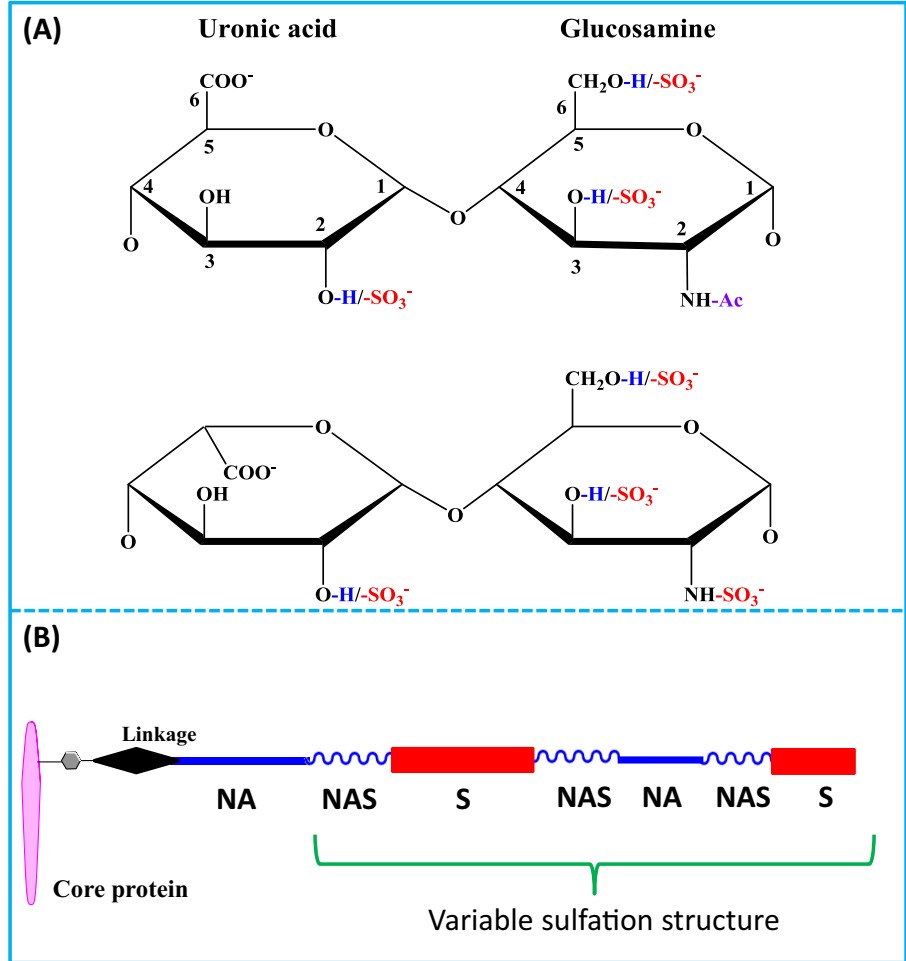

**Figure 4 Structures of disaccharide units of HS and heparin.** (A) Structure of disaccharide unit of heparin/HS. Top: the glucuronic acid containing disaccharide. This is generally not or only slightly modified by sulfation (in red). Bottom: the iduronic acid containing disaccharide, which always contains an N-sulfated glucosamine (red) and is often further modified by O-sulfation (red). (B) Structure of HS chains. The polysaccharide chain is covalently linked to a serine on the proteoglycan core protein. The sulfate groups are added by sulfotransferases after the GAG chain is polymerised. Due to the hierachical dependence of the post polymerisation reactions and the sulfation of discrete blocks of N-acetylglucosamines by N-deacetylase-N-sulfotransferases (NDSTs), the HS chain has a domain structure of alternating NA (GlcA/GlcNAc), NAS (~one disaccharide in two is N-sulfated) and S (every glucosamine is N-sulfated) domains. Chain lengths vary from ~25 disaccharides to over 100. Heparin, a common experimental proxy for heparan sulfate is ~30 disaccharides in length and can be considered to be a highly sulfated NS domain.

units, (4-GlcA $\beta$1–4 GlcNac$\beta$1-)$_n$ (where n ~25 to 100) are then added by the copolymerases EXT1 and EXT2 (*Dreyfuss et al., 2009*; *Lin, 2004*; *Tumova, Woods & Couchman, 2000*). After the synthesis of the chain, clusters of N-acetyl glucosamine are removed and N-sulfate groups are added by the dual activity N-deacetylase-N-sulfotransferases (NDSTs) (*Dreyfuss et al., 2009*; *Lin, 2004*; *Tumova, Woods & Couchman, 2000*). The subsequent modifications are on N-sulfated glucosamine containing disaccharides or their neighbours: an epimerase converts

glucuronic acid to iduronic acid, which may then be 2-O sulfated and the glucosamine may be 6-O and 3-O sulfated (*Dreyfuss et al., 2009*; *Lin, 2004*; *Tumova, Woods & Couchman, 2000*).

Since NDSTs selectively act on blocks of disaccharides, the modified heparan sulfate has a domain structure of NA, NAS domain and S domains (Fig. 4B) (*Connell & Lortat-Jacob, 2013*; *Dreyfuss et al., 2009*; *Gallagher, 2015*; *Murphy et al., 2004*; *Ori, Wilkinson & Fernig, 2008*). Differences in sulfation level of the NAS and NS domains provide the means for heparan sulfate to bind with varying degrees of selectivity to over 435 proteins (*Ori, Wilkinson & Fernig, 2008*; *Xu & Esko, 2014*), including FGFs binding S-domains and antithrombin III binding transition domains (*Turnbull, Powell & Guimond, 2001*; *Xu & Esko, 2014*). Since the modification reactions by the sulfotransferases do not go to completion, the length and level of sulfation of heparan sulfate chains are also variable in different cells and extracellular matrices (*Dreyfuss et al., 2009*; *Kirkpatrick & Selleck, 2007*; *Ori, Wilkinson & Fernig, 2008*; *Xu & Esko, 2014*).

## FGFR

FGFRs, spanning the membrane, are the key to transferring induced signals into the cell, which direct the target cell activities, such as cell proliferation, differentiation and migration (*Beenken & Mohammadi, 2009*; *Ornitz, 2000*; *Turner & Grose, 2010*). Five different FGFRs (FGFR1–4 and FGFRL1) and many of their alternative spliced isoforms have been found to bind with FGFs and activate a large number of signalling pathways (*Dorey & Amaya, 2010*; *Itoh & Ornitz, 2011*; *Turner & Grose, 2010*; *Wiedemann & Trueb, 2000*). FGFR1–4 possess three extracellular immunoglobulin-like loops, I, II and III (often termed D1, D2 and D3), a transmembrane linker and a cytoplasmic kinase domain (*Beenken & Mohammadi, 2009*; *Dorey & Amaya, 2010*; *Goetz & Mohammadi, 2013*; *Turner & Grose, 2010*). FGFRL1 differs in that its intracellular domain lacks a tyrosine kinase (*Kim et al., 2001*; *Sleeman et al., 2001*; *Wiedemann & Trueb, 2000*). Half of D3 is encoded in FGFR1, FGFR2 and FGFR3 by alternative exons. This gives rise to the 'b' and 'c' isoforms of the transmembrane receptor, which impart additional ligand selectivity (*Ornitz et al., 1996*; *Zhang et al., 2006*). In addition, the tyrosine kinase FGFRs also bind heparan sulfate (*Kan et al., 1993*; *Powell, Fernig & Turnbull, 2002*), which leads to one ternary FGF-FGFR-heparan sulfate signalling structure (*Schlessinger et al., 2000*).

The FGFRs have varying degrees of selectivity for different FGFs, and the selectivity is most conserved between FGFs in the same subfamily (*Ornitz et al., 1996*; *Xu et al., 2013*; *Zhang et al., 2006*). FGF1 was recognised as a universal ligand for all the FGFRs, while FGF2 and members of the FGF4 subfamily prefer to interact with FGFR 1c (*Zhang et al., 2006*). For the 'c' isoform, the preference is FGFR 1c > FGFR 2c and FGFR 3c, though the FGF4 subfamily ligands are clearly distinguished from the FGF1 subfamily in terms of their selectivity for FGFR 1b, which they do not bind, in contrast to FGF1 and FGF2 (*Ornitz et al., 1996*; *Zhang et al., 2006*). Members of the FGF8 and FGF9 subfamilies preferentially bind to FGFR 3c (FGFR 3c > FGFR 2c and 1c), while members of FGF7 subfamily mainly bind FGFR 2b and 1b (*Ornitz et al., 1996*; *Zhang et al., 2006*).
## KLOTHO CO-RECEPTORS

Klotho co-receptors (alpha and beta-Klotho/KLB) are type 1 transmembrane proteins that define tissue specific activities of circulating endocrine FGFs (for reviews see *Belov & Mohammadi, 2013*; *Kuro-o, 2012*; *Martin, David & Quarles, 2012*). α*Klotho* was originally identified as an aging suppressor gene (*Kuro-o et al., 1997*). Findings that mice with disrupted α*Klotho* expression displayed identical phenotypes to mice deficient in FGF23, including shortened life span, growth retardation, muscle atrophy, vascular calcification in the kidneys and disrupted serum phosphate balance, led to the discovery of αKlotho as an obligatory co-receptor for FGF23 to bind and activate FGFR in the kidney (*Kurosu et al., 2006*; *Urakawa et al., 2006*). Bone derived FGF23 acts in the *aKlotho* expressing kidney to regulate vitamin D and phosphate homeostasis. Beta-Klotho (KLB) was identified by its sequence homology to αKlotho (*Ito et al., 2000*), and later identified as a co-receptor to allow FGF19 and −21 to bind and signal via their canonical FGFRs in bile acid, glucose and lipid metabolism, respectively (*Kharitonenkov et al., 2008*; *Kurosu et al., 2007*; *Lin et al., 2007*; *Wu et al., 2007*). The extracellular domains of Klotho co-receptors are composed of two KL domains with sequence homology to beta-glucosidases (*Ito et al., 2000*; *Kuro-o et al., 1997*). αKlotho also exists in a secreted form, either via alternative splicing or via shedding of the extracellular domain by matrix metalloproteases. The secreted form of Klotho has been shown to modulate glycans on Transient Receptor Potential calcium channels TRPV5 and TRPV6 (*Chang et al., 2005*) and renal outer medullary potassium channels (ROMK1) (*Cha et al., 2009*) *in vitro*, increasing their cell-surface retention.

## ASSEMBLY OF SIGNALLING COMPLEXES

The binding of the FGF ligand to its receptor with/without heparan sulfate (co-receptor) causes the FGFR to dimerise. This in return enables phosphorylation of tyrosine residues in the kinase activation loop and then of tyrosines that are docking sites for signalling proteins (*Goetz & Mohammadi, 2013*). The latter activate most intracellular signalling pathways, *e.g.* RAS-RAF-MAPK and PI3K-AKT, which regulate cell fate and specific cell activities (*Dorey & Amaya, 2010*; *Turner & Grose, 2010*). Previous studies suggest heparan sulfate (or its experimental proxy heparin) is required for many, but not all signalling (*Izvolsky et al., 2003*). FGF signalling can be negatively regulated by internalisation and degradation, as well as by transmembrane regulators, such as FGFRL1, and intracellular ones, *e.g.* Sprouty and Spred (*Casci, Vinos & Freeman, 1999*; *Hacohen et al., 1998*) and MAPK phosphatase 3 (*Turner & Grose, 2010*). Since there is a great diversity of FGF ligands, FGFR isoforms and heparan sulfate structure and feedback loops, the understanding of FGF signalling is still far from complete (*Dorey & Amaya, 2010*).

## ALTERNATIVE PARTNERS

FGFs and FGFRs interact directly with a large number of other partners, both extracellularly and, following the internalisation of ligand-receptor complexes, intracellularly. In some instances, e.g., FGF-2 binding integrins (*Rusnati et al., 1997*),

these may be additional to the core complex of FGF, FGFR and heparan sulfate, but in other cases, e.g., cadherins, these are orthogonal partners of one component of the core FGF communication system, the FGFR (*Doherty & Walsh, 1996*). A partial list of the alternative extracellular partners has been reviewed (*Polanska, Fernig & Kinnunen, 2009b*). The intracellular partners and functions of FGF receptor-ligand complex components translocated to the nucleocytoplasmic space have also been recently reviewed (*Coleman et al., 2014*).

## DIVERSIFICATION AND SWITCHING OF FUNCTION: HINTS FROM *C. ELEGANS*

The functions of FGFs in mammals are very diverse, which reflects the expansion at the molecular level of the FGF communication system that accompanied the evolution of more complex animal body plans and physiology. In contrast *C. elegans* possesses one of the simplest FGF communication systems, comprising two ligands, EGL-17 (*Burdine et al., 1997*) and LET-756 (*Roubin et al., 1999*), a single FGFR, EGL-15 (*DeVore, Horvitz & Stern, 1995*), and two orthologues of Klotho, KLO-1 and KLO-2 (*Polanska et al., 2011*). The EGL-15 receptor is alternatively spliced into an "A" and a "B" isoform, resulting in structural differences in the extracellular domain of the receptor between immunoglobulin domains I and II (*Goodman et al., 2003*). Work in *C. elegans* provides an insight into the relation between the paracrine and endocrine activities of FGFs and how heparan sulfate binding of FGFs may have changed during the expansion of the family. This in turn provides one line of evidence to support the argument that we can manipulate the FGF communication system therapeutically for patient benefit in repair and metabolic scenarios, without undue risk of tumourigenesis.

The major functions of EGL-15 are paracrine in the cell migration of sex myoblasts (*DeVore, Horvitz & Stern, 1995*), neural development (*Bulow, Boulin & Hobert, 2004*; *Fleming, Wolf & Garriga, 2005*), and an early essential function (*DeVore, Horvitz & Stern, 1995*) associated with physiological homeostasis (*Huang & Stern, 2004*; *Polanska et al., 2009a*; *Polanska et al., 2011*). EGL-17/FGF acts as a chemoattractant to guide sex myoblasts (*Burdine, Branda & Stern, 1998*), whereas LET-756/FGF is required for the essential function of EGL-15, as animals lacking LET-756 arrest at early larval stage (*Roubin et al., 1999*). In mammals, the "IIIb" and "IIIc" isoforms of FGFRs enable ligand selectivity. In *C. elegans* this ligand to receptor pairing is determined in part by tissue specific expression of the ligand and the "A" and "B" receptor isoforms (*Goodman et al., 2003*; *Lo et al., 2008*). *egl-15(5B)* is predominantly expressed in the hypodermis (*Lo et al., 2008*), where it mediates fluid homeostasis (*Huang & Stern, 2004*), whereas *egl-15(5A)* isoform is expressed in the M lineage, which gives rise to the sex myoblasts. Heterologous expression of *egl-17*, driven by the *let-756* promoter, can stimulate EGL-15(5B) and partially rescue the larval arrest phenotype of mutants lacking LET-756, and expression of *let-756* driven by *egl-17* promoter can partially rescue sex myoblast migration in EGL-15-deficient worms (*Goodman et al., 2003*). However, although expression of either isoform of *egl-15* in the hypodermis can mediate the fluid

homeostasis phenotype, only EGL-15(5A) isoform can mediate sex myoblast chemoattraction (*Lo et al., 2008*). Thus, the functional specificity of EGL-15 is determined by the extracellular receptor isoform and the availability of the ligand. That there are no multicellular organisms possessing just a single FGF ligand and one FGFR isoform may reflect the importance of selective communication between tissue compartments and of the ability of cells to switch the ligand channel they are tuned to during development, without changing the receiver (the receptor kinase and downstream signalling).

The role of EGL-15 in the regulation of *C. elegans* fluid homeostasis was first discovered in mutants of a phosphatase, which acts downstream of EGL-15 (*Kokel et al., 1998*). This phosphatase, CLR-1, acts as a negative regulator of EGL-15, and its absence leads to excess EGL-15 activity and accumulation of fluid within the *C. elegans* pseudocoelom and a clear (clr) phenotype. Excess EGL-15 activity and clear phenotype can also be achieved by mutation of *N*-glycosylation sites in the extracellular domain of the EGL-15 receptor (*Polanska et al., 2009a*), as *N*-glycans act as a brake on receptor activation (*Duchesne et al., 2006*). Under laboratory conditions *C. elegans* must actively excrete fluid. The major organs responsible for fluid balance are the hypodermis, which expresses *egl-15(5B)* and *klo-2* (*Polanska et al., 2011*) and the excretory canal, which is equivalent to the mammalian kidney and expresses *klo-1* (*Polanska et al., 2011*). Complete loss of function of EGL-15 or LET-756 leads to loss of *klo-1* expression and lack of functional excretory canals (*Polanska et al., 2011*), a likely explanation of the early larval lethality of the mutants defective of LET-756/EGL-15 signalling.

Thus, in *C. elegans* the same FGFs act as growth factors, morphogens and hormones, whereas in mammals different FGFs perform the local and systemic functions. EGL-15 associates with Klotho co-receptors to mediate the fluid homeostasis function (*Polanska et al., 2011*), which is entirely analogous to the mode of action of endocrine FGFs in mammals. Although there is currently no genetic evidence to suggest that, as in mammals, the assembly of a signalling complex of the *C. elegans* FGF ligands with EGL-15 and subsequent receptor activation would depend on the heparan sulfate co-receptor *in vivo*, biochemical evidence shows that EGL-15/FGFR binds to heparin, a proxy for heparan sulfate (*Polanska et al., 2011*), whereas sequence alignment of EGF-17 and LET-756 to mammalian FGFs indicates that they possess heparan sulfate binding sites (*Xu et al., 2012*). Importantly, binding to heparan sulfate would not preclude a hormone homeostatic activity of *C. elegans* FGFs, since the range of the FGF would be significant compared to the animal's body size; *C. elegans* is small, (adult hermaphrodites ~1 mm). Thus, a reasonable hypothesis is that the communication system used in development, LET-756 and EGL-15, is then co-opted into endocrine homeostasis. As animals grew larger, this would no longer be possible. Diversification of the FGF family and weakening of heparan sulfate binding would then allow both the growth factor/morphogen activity, which is local due to heparan sulfate binding and the systemic hormonal activity to be retained. In support of this idea is the demonstration that a human FGF-1 with its primary heparan sulfate binding site mutated is reprogrammed from a growth factor to a FGF-21 like hormone, controlling metabolism (*Suh et al., 2014*). A corollary is that the

interaction of paracrine FGF ligands with heparan sulfate is one key to understanding their function, their roles in disease and hence their therapeutic potential.

## THE ASPECTS OF FGF ACTIVITIES LINKED TO CANCERS

As noted above, a great deal of the early work on FGFs was from a cancer perspective, though there was also a considerable effort directed at regeneration of damaged tissues. While there is a bias in the scientific literature against reporting negative results, there are some reports that showed in animal models and in clinical samples that there was not a simple relationship between FGF ligand expression and tumour formation and progression. Thus, when FGF-2 mRNA levels were analysed in a cohort of breast cancer patients, elevated expression of FGF-2 mRNA correlated with a good prognostic outcome, the opposite of the result expected from the naïve perspective that "FGF2 = uncontrolled growth + angiogenesis = cancer" (*Anandappa et al., 1994*). Similarly, in a syngeneic rat model of breast cancer, overexpression of FGF2 failed to produce any metastases (*Davies et al., 1996*). Given the difficulty in publishing negative results, there is likely a very large body of work that demonstrates the absence of a direct association between the expression of FGF ligands and cancer.

One reason is that, at least for FGF-1 and FGF-2, the ligand is often not limiting. That is, there is a lot of ligand stored on heparan sulfate in tissues, which is then accessed during repair. The discovery of the storage of FGF2 on heparan sulfate of extracellular matrix (*Vlodavsky et al., 1987*) was followed by the realisation that stored FGF2 could elicit a response at least in cultured cells (*Presta et al., 1989*) by diffusion within matrix (*Duchesne et al., 2012*). The expression of other FGF ligands is, in contrast, often induced. However, like FGF1 and FGF2, their activity is restricted, again through binding to heparan sulfate and due to their selectivity for FGFRs. An important facet of development and endogenous tissue repair is the mobilisation of FGF ligands by heparanase, a beta glucuronidase, which cleaves heparan sulfate in NA and NAS domains, liberating growth factor bound to an S domain (*Arvatz et al., 2011*; *Barash et al., 2010*; *Kato et al., 1998*; *Patel et al., 2010*; *Ramani et al., 2013*). This plays a key role in many cancers (*Arvatz et al., 2011*; *Barash et al., 2010*; *Ramani et al., 2013*). Therefore, the mechanistic link between the FGF communication system and cancers is on the side of the mobilisation of FGFs from such stores (particularly by heparanase, though proteases are likely to also have a role) and of increases in the activity of FGFRs (*Carter, Fearon & Grose, 2015*; *Turner & Grose, 2010*). Thus, in contrast to their ligands, the FGFRs are established drivers of tumour progression. This arises from: activating mutations; isoform switching, e.g., between the classic epithelial, FGFR2-IIIb isoform that binds FGF-7 family members and FGFR2-IIIc isoform, that binds epithelial synthesized and mesenchymally stored FGFs, including FGF-2 (*Carter, Fearon & Grose, 2015*; *Turner & Grose, 2010*).

## FGFS AS REPAIR FACTORS

The use of FGFs to repair damaged tissue is a long-standing research area, however, until recently in the West it was entirely confined to model systems. In Japan, alongside the
cancer research track, a repair track leading to clinical applications was developed. In contrast, in China FGF research was from the late 1980s spearheaded by the drive to develop a biotechnology industry. This resulted in successful engineered production of FGF1 and FGF2, (e.g., *Wu et al., 2005*; *Wu et al., 2004*; *Yao et al., 2006*; *Zhao et al., 2004*) and a substantial effort in experimental medicine, including pharmacokinetic and toxicity studies, (e.g., *Li et al., 2002*; *Xu et al., 2003*) to develop clinical applications. In much of this work the original nomenclature, aFGF and bFGF is used for FGF1 and FGF2, respectively; in the following summary of some of the clinical studies, the currently accepted numerical nomenclature is employed. A major clinical focus in China has been the use of FGF2 as a repair/regeneration factor in conditions as diverse as burns, chronic wounds, oral ulcers, vascular ulcers, diabetic ulcers, pressure ulcers and surgical incisions. As the Chinese studies are not generally accessible, we have summarised a number of these below, alongside other work on similar conditions from Japanese research groups and the few Western clinical trials. The very extensive preclinical literature is not covered.

## THERAPEUTIC APPLICATIONS OF FGF2

FGFs have been investigated as therapeutic agents in a number of diseases, with varying success. We outline some of the studies (Table 1), ranging from case series and observational studies carried out prospectively (before treatment has been initiated) or retrospectively (after treatment has been completed) or a combination of retrospective and prospective approaches to well-designed randomised controlled studies carried out prospectively. The quality of the studies is variable and with details of the FGFs used not available in all instances (Table 1).

### Burns

*Liu, Jiang & Tan (2005)* investigated the use of FGF2 in the treatment of burns and chronic wounds. Patients were divided into a burn wound group (n = 62), a donor site wound group (n = 36) and a chronic wound group (n = 65). The burn wounds included superficial partial thickness burns and deep partial thickness burns; chronic wounds included wounds that did not heal following routine treatment for 4 weeks, residual granulation wounds, pressure ulcers, sinuses, and diabetic ulcers. The burn wound group was treated with FGF2 in addition to the standard treatment. Self-control randomization was applied to the burn wound group and donor site wound group, with comparisons of the same subject before and after treatment. The control group was treated with equal amounts of saline in addition to the standard treatment. The results showed that FGF2 significantly shortened the time to complete wound healing in the three wound groups compared to the control group.

*Guo (2006)* randomly assigned 80 cases of deep partial thickness burn wounds to a treatment group and a control group. In the treatment group, a gauze pad impregnated with FGF2 solution was applied to the debrided wound, which was then covered with another gauze pad containing 1% (w/w) silver sulfadiazine. Apart from substituting normal saline for FGF2, the control group was subjected to the same treatment as the FGF2 group. The results showed that the average healing time for superficial partial

**Table 1 Therapeutic applications of FGFs.** Summary of the clinical uses of FGFs and the types of study.

| FGF | Disease/condition | Author, year | Type of study | FGF preparation/ concentration | Outcome |
|---|---|---|---|---|---|
| FGF2 | Burns and chronic wounds | Liu, Jiang & Tan, 2005 | Prospective, self controlled randomisation study | FGF2 soaked gauze (20,000 AU/100 cm$^2$) | Healing time was significantly reduced in the FGF2 treated groups (burns and chronic wounds) compared to the control group. |
| FGF2 | Burns | Guo, 2006 | Randomised controlled study | FGF2 soaked gauze (20,000 AU/100 cm$^2$) | Healing time was significantly reduced in the FGF2 treated group. |
| FGF2 | Burns (second degree) | Akita et al., 2008 | Randomised controlled study | FGF2 Spray (30 mg/30 cm$^2$ area) | Healing time was significantly reduced and quality of scar improved in the FGF2 treated group. |
| FGF2 | Sutured wounds (following skin tumour removal) | Ono et al., 2007 | Prospective, non-randomised case control study | Intradermal FGF2 injections (low dose −0.1 mg FGF2 per 1 cm of wound, high dose −1.0 mg FGF2 per 1 cm wound) and high FGF2 rinses (0.1 mL of 10 mg/mL FGF2 solution per 1 cm wound) | Scarring was significantly reduced in the FGF2 treated groups (low and high doses of FGF2). |
| FGF2 | Donor sites (split thickness skin grafts) | Xu, Li & Fan, 2000 | Randomised self-controlled trial | FGF2 soaked gauze (150 U/cm$^2$ for the first 3 days followed 100 U/cm$^2$ subsequently) | FGF2 significantly reduced healing time and improved quality of the scar in the treatment group. |
| FGF2 | Avulsion wounds/ full-thickness skin graft | Matsumine, 2015 | Prospective, case series | FGF2 spray (1 μg/cm$^2$ of graft bed) | FGF2 application resulted in wound healing with flexible scars in all cases. |
| FGF2 | Sutured wounds (cosmetic surgery) | Lu, Jin & Pang, 2006 | Observational study | FGF2 soaked gauze (concentration details not available) | FGF2 application resulted in a significantly shorter healing time and better quality of scar. |
| FGF2 | Wound dehiscence following Caesarean section | Chen & He, 2004 | Randomised controlled study | FGF2 spray (2–4 mL per application; details of concentration not available) | FGF2 resulted in a significantly shorter healing time in wounds <5 cm in the treatment group. |
| FGF2 | Tibial shaft fractures | Kawaguchi et al., 2010 | Randomised, double blind, placebo-controlled study | 2 percutaneous injections of hydrogel (0.5 mL each, containing 0, 0.4, or 1.2 mg of FGF2) | FGF2 accelerated healing of tibial fractures in the treatment groups. |
| FGF2 | Traumatic skin ulcers | Zang, Zha & Yao, 2005 | Randomised controlled study | FGF2 biological protein sponge (concentration details not available) | FGF2 application resulted in a significantly higher healing rate in the treatment group. |
| FGF2 | Recurrent aphthous stomatitis | Jiang et al., 2013 | Double blind, randomised controlled trial | Paste A contained Diosmectite (DS) −80 mg/g and FGF2 −10 mg/g. Paste C (FGF2 paste) primarily contained FGF2 (10 mg/g) | Paste A (DS + FGF2) significantly reduced ulcer pain scores and ulcer size. |

| FGF | Disease/condition | Author, year | Type of study | FGF preparation/ concentration | Outcome |
|-----|-------------------|--------------|---------------|--------------------------------|---------|
| FGF2 | Periodontal regeneration | Kitamura et al., 2011 | Double blind, randomised controlled trial | 0.2%, 0.3%, or 0.4% FGF2 gel for local application | The periodontal fill was significantly higher in the FGF2 treated group. |
| FGF2 | Aphthous ulcers | Ren & Shun, 2002 | Randomised, double-blinded, controlled trial | FGF2 spray (300 AU/application, 4 times/day) | FGF2 significantly reduced the ulcer healing time in the treatment group. |
| FGF7 | Oral mucositis (Chemo-radiotherapy) | Goldberg et al., 2013 | Retrospective observational study | Three daily doses of FGF7 (60 $\mu$g/kg/day) were given prior to transplant admission with the third dose given no fewer than 24 hours prior to administration of chemotherapy or radiotherapy. Six hours after stem cell infusion, patients received three further daily doses of FGF7 (60 $\mu$g/kg/day). | FGF7 significantly reduced the number of days of total parenteral nutrition, patient-controlled analgesia and length of hospital stay in patients receiving total body irradiation. |
| FGF2 | Traumatic perforations of the tympanic membrane | Lou and Wang, 2013 | Prospective, sequential allocation, three-armed, controlled clinical study | 0.25 mL (4–5 drops) of FGF2 (21,000 IU/5 mL) solution | Average closure time was significantly shorter in the FGF2 application group. |
| FGF2 | Pressure ulcers | Robson et al., 1992 | Randomised, blinded, placebo-controlled trial | FGF spray (concentrations of 100 $\mu$g/mL, 500 $\mu$g/mL, or 1000 $\mu$g/mL | FGF2 resulted in a significantly higher number of patients with 70% decrease in size of the ulcer in the FGF2 treated group. |
| FGF2 | Diabetic ulcer | Uchi et al., 2009 | Randomised, double blinded, dose-ranging, placebo-controlled trial | FGF2 solution (0.01% and 0.001% w/v) | Cure rates were significantly higher in the 0.01% w/v FGF2 treated group. |
| FGF2 | Critical limb ischaemia | Kumagai et al., 2015 | Phase I-IIa trial | 200 $\mu$g of FGF2 incorporated gelatin hydrogel microspheres injected intramuscularly into the ischemic limb | Transcutaneous pressure, distance walked in 6 minutes, rest pain scale and cyanotic pain scale showed significant improvement at 24 weeks post-treatment with FGF2 as compared to pre-treatment. |

thickness burn wounds in the FGF2-treated group was significantly shorter as compared to the control group ($9.51 \pm 1.86$ days *vs.* $12.43 \pm 2.03$ days, $p < 0.05$). Similarly, the healing time in the deep partial thickness burn wounds in the FGF2 treatment group was significantly shorter than the control group ($18.36 \pm 4.87$ days *vs.* $22.35 \pm 5.60$ days, $p < 0.01$).

FGF2 has also been shown to accelerate healing and improves scar quality in second-degree burns (Akita et al., 2008). Since the speed of wound healing is an important factor influencing the outcome of treatment, as well as a crucial step in burn wound treatment, and the quality of wound healing has a direct bearing on the quality of life of patients, FGF2 clearly has clinical efficacy in a variety of burn settings.

## Surgical wounds

### Surgical incisions

Surgical incisions leave scars as part of the normal healing process. These scars vary from being narrow, wide, atrophied or hypertrophic and sometimes cause medical problems, or social ones, because of their cosmetic appearance (*Rockwell, Cohen & Ehrlich, 1989*). A study by *Ono et al. (2007)* examined the effect of local administration of FGF2 on sutured wounds. FGF2 was injected into the dermis of the wound margins using a needle immediately after the skin was sutured following an operation. None of the patients treated with FGF2 had hypertrophic scars compared to the control group and scarring was significantly lower in the groups treated with FGF2, as compared to the control group.

### Skin graft wounds

Healing of the donor site wounds, created after skin graft harvesting, involves the regeneration of epithelial cells in the residual skin appendages (*Metcalfe & Ferguson, 2007*). Early healing of donor site wounds helps to reduce trauma, thereby facilitating the treatment of the primary disease.

*Xu, Li & Fan (2000)* conducted a clinical study to examine the efficacy of topical application of FGF2 on 48 donor site wounds in 34 patients, which were created by harvesting intermediate split thickness skin grafts. The wounds before treatment served as self-controls. Following the harvesting of the skin grafts, the wound surface was evenly coated with FGF2 using a cotton swab, covered with vaseline gauze, and dressed. The control wounds were smeared only with the vehicle without FGF2, the rest of the topical treatment procedures being identical as the treatment group. The results showed healing time in wounds treated with FGF2 was 2.8 days shorter compared to control wounds ($p < 0.01$). Moreover, FGF2-treated wounds appeared flatter, smoother and firmer and were difficult to tear off, as compared to the control wounds. The use of FGF2 yielded no adverse reactions.

### Full thickness skin grafts in avulsion injuries

*Matsumine (2015)* described the topical use of FGF2 in the treatment of avulsion wounds (as a result of skin and/or underlying tissue torn away due to trauma) with full thickness grafts using the avulsed skin. The contaminated subcutaneous fat tissue on the inside of skin was excised and the avulsed skin was processed into a full-thickness skin graft. Drainage holes (5–10 mm in diameter) were made on the graft to prevent seroma and haematoma formation. FGF2 was sprayed onto the graft bed, followed by application of the graft. Skin grafts that did not take were scraped away, preserving the revascularized viable dermis where possible. FGF2 was then sprayed again onto this surface to promote epithelialization (proliferation of epithelial cells to cover the wound). Wound closure was achieved in all cases with conservative therapy. This procedure promoted wound healing with the formation of good-quality, flexible scars and prevented postoperative ulcer formation and scar contracture.

### Cosmetic surgical incisions

Wound healing quality is important in the success of cosmetic surgery. *Lu, Jin & Pang (2006)* examined the effects of FGF2 on wound repair in 60 female patients who underwent cosmetic surgery. All surgical incisions were clean cuts, and self-controls (another incisional wound on the same patient) were used. In the treatment group FGF2 was applied once daily until removal of stitches, starting with the first postoperative day. Wounds due to laser resurfacing were smeared with FGF2 twice daily until natural decrustation occurred. The control group was subjected to conventional dressing change until removal of stitches. The results showed that in the FGF2 group whose wounds resulted from laser resurfacing, the average decrustation time was significantly shorter than in the control group (6.2 days *vs.* 8.1 days, p < 0.05). The FGF2-treated groups showed good healing. In addition, exudate and swelling post surgery were milder in the FGF2 groups than in the control group. There were no adverse reactions in the FGF2 groups. Quality of wound healing was superior and the healing time was shorter in the FGF2 groups as compared to the control group, indicating that FGF2 has a favourable effect on cosmetic surgical incision healing.

### Obstetric wounds

Dehiscence of caesarean section incisions may occur in the form of a superficial dehiscence, in which the skin and subcutaneous fat layer break open, most often due to fat liquefaction caused by subcutaneous fat hypertrophy in pregnant women. In addition, a long trial of labour, excessive vaginal examinations, vaginitis, and intrauterine infections may potentially lead to an increase in infected incisions. Anaemia, hypoproteinemia, malnutrition, and diabetes in the perinatal period can result in poor healing capacities of local tissues. These factors can adversely affect wound healing extending hospital stay, and increasing costs.

*Chen & He (2004)* randomly assigned 60 patients with wound dehiscence following a caesarian section to two groups: an observation group and a control group. After debridement of the wounds, FGF2 was sprayed on the wounds and they were sutured the next day. Wound dressings were changed regularly. The control group was treated similarly, but without the use of FGF2 spray. Healing time was significantly shorter in patients with dehiscence measuring 5 cm in size or below treated with FGF2 compared to the control group (6.8 ± 1.5 days *vs.* 11.2 ± 1.2, *p* < 0.01). In contrast, in patients with dehiscence measuring 5 cm and above secondary suturing was undertaken. In this instance there was no significant difference in the FGF2 treated group compared to their respective control group (7.6 ± 1.0 days *vs.* to 7.4 ± 0.8, *p* > 0.05); it is likely that this control group's shortened healing time, compared to the control group with dehiscence measuring 5 cm or less, was due to secondary suturing.

### Orthopaedic trauma wounds

FGF signalling plays an important role in skeletal development (*Su, Du & Chen, 2008*). Tissue necrosis and infection of fresh skin defect wounds, grafted flaps, and skin grafts occurs following orthopaedic trauma surgery. In order to shorten the healing time and

reduce the rate of skin re-grafting, FGF2 has been directly applied to fresh and debrided necrotic wounds.

*Kawaguchi et al. (2010)* conducted a randomised, placebo-controlled trial, investigating the direct application of FGF2 in a gelatin hydrogel on traumatic tibial fractures. A single injection of gelation containing placebo or low dose FGF (0.8 mg) or high dose FGF (2.4 mg) was administered into the fracture gap at the end of intramedullary nailing surgery. Radiographic bone union was significantly higher in the FGF2 treated groups, with no significant difference between the two FGF2 dosage groups.

*Zang, Zha & Yao (2005)* investigated the use of a FGF2 biological protein sponge for traumatic ulcers. A sterile FGF2 biological protein sponge was applied to traumatic skin ulcers in 20 patients. The results showed that the wound-healing rate within 3 weeks was 95% in the FGF group and 55% in the control group, and that the rate of skin re-grafting in the FGF2 group was significantly lower than that in the control group. Wound secretions and peri-wound inflammation were markedly less severe in the FGF2 group as compared to the control group. No obvious adverse reactions were reported in either group. These data indicate that FGF2 biological protein sponges may promote the healing of traumatic ulcers and shorten healing time.

### Oral diseases

Oral ulcers are a common disease of the oral mucosa and tend to recur. Pathologically, ulcers of oral mucosa are mainly characterized by dissolution, rupture, and shedding of local oral mucosal epithelium to form non-specific ulcers.

A study by *Jiang et al. (2013)* investigated the use of topical application of diosmectite (DS; an insoluble silicate) and FGF2 paste in the treatment of minor recurrent aphthous stomatitis (repeated formation of benign, non-infectious ulcers in the mouth). Four pastes, containing FGF2 and DS, DS alone, FGF2 alone, and vehicle only, were used in 129 participants. DS-FGF2 significantly lowered ulcer pain scores ($p < 0.05$ for days 3, 4, 5, and 6) as compared to the other pastes. Ulcer size was significantly reduced ($p < 0.05$ for days 2, 4, and 6) in this group. No obvious adverse drug effects were observed.

*Kitamura et al. (2011)* conducted a multicentre, randomised, double blind, placebo-controlled trial, in accordance with Good Clinical Practice guidelines, to clarify the efficacy and safety of FGF2 use in periodontal regeneration. The percentage of bone fill was significantly higher in the FGF2 treatment group as compared to the 'vehicle alone' group at 36 weeks. Also, there were no serious adverse effects in the treatment group.

Radiotherapy is commonly used to treat head and neck cancer. However, when the radiation dose rises to about 20 Gy–30 Gy, acute inflammation of the oral mucosa usually occurs, the symptoms of which include, among other things, oropharyngeal pain, and oral ulcers associated with oedema or pseudomembrane formation. Food intake is affected as a result. Moreover, the severity of the symptoms increases with the radiation dose. Patients who experience serious symptoms have to suspend the treatment, and the final efficacy of the treatment is thus impaired (*Vera-Llonch et al., 2006*; *Worthington et al., 2011*).

Myeloablative allogeneic haemopoietic stem cell transplantation is an established treatment for haematologic malignancies and oral mucositis is a known complication

arising from high dose chemotherapy and radiation therapy. *Goldberg et al. (2013)* performed a retrospective study investigating the use of peritransplant Palifermin (recombinant FGF7) and found that it significantly reduced the number of days of total parenteral nutrition, patient-controlled analgesia and length of hospital stay in patients receiving total body irradiation as compared to those receiving chemotherapy based transplantation.

*Ren & Shun (2002)* conducted a double-blind study, in which 121 patients with mild aphthous ulcers (mouth ulcers) were randomly assigned to either a FGF2 group (n = 63) or a control group (n = 58). In the FGF2 group, FGF2 was locally sprayed onto the surface of ulcers; in the control group, 0.2% (w/v) chlorhexidine solution was sprayed on the ulcers. The results showed that the effective rate at day 3 was 90.48% in the FGF2 group and 60.34% in the control group (p < 0.05). Meanwhile, the average healing time of ulcers was significantly shorter in the FGF2 group than in the control group (Chi square test; *p* < 0.05). The results show that FGF2 exhibits significant efficacy for mild recurrent aphthous oral ulcers.

### Tympanic membrane perforations

While most traumatic perforations of the tympanic membrane tend to heal spontaneously, large perforations may often fail to do so. The management of these is still open to debate, with a number of specialists recommending an early myringoplasty to improve outcomes (*Conoyer, Kaylie & Jackson, 2007*). *Lou & Wang (2013)* undertook a prospective, sequential allocation, three-armed, controlled clinical study to compare perforation edge approximation *vs.* FGF2 application in the management of traumatic perforations of the tympanic membrane. Patients were divided into 3 groups: no intervention (n = 18), edge approximation (n = 20) and direct application of FGF2 (n = 20). Otoscopy was performed before and after treatment and response measurements were made, such as closure rate, closure time and rate of otorrhoea. Perforation closure was significantly higher in the FGF2 group (100%) as compared to the edge approximation (60%) and control (56%) groups (p < 0.05). Average closure time was significantly shorter in the FGF2 treatment group (12.4 ± 3.6 days), as compared to the edge approximation (46.3 ± 8.7 days) and control (48.2 ± 5.3 days) groups (p < 0.05). *Lou, Wang & Yu (2014)* showed that a lower dose (0.1 to 0.15 mL) of FGF2 (21,000 IU/5 mL) was more effective than a higher dose (0.25 to 0.3 mL). *Hakuba et al. (2010)* demonstrated that FGF2 combined with atelocollagen was an effective treatment for chronic tympanic membrane perforations.

### Pressure ulcers

Treatment of pressure ulcers is a major problem for clinical care. Pressure ulcers can increase patients' suffering, extend the duration of illness, and, when serious, may even prove to be life threatening due to sepsis resulted from secondary infection. Commonly used treatments over the years have included innovative mattresses, ointments, creams, solutions, dressings, ultrasonography, ultraviolet heat lamps, and surgery.

*Robson et al. (1992)* investigated the role of FGF2 in the treatment of pressure ulcers with a randomized, blinded, placebo-controlled trial, which enrolled 50 patients with

pressure ulcers varying in size from 10 to 200 cm$^3$. The results showed that, compared with placebo-treated patients, the number of FGF2-treated patients whose ulcers shrank by 70% increased significantly (60/100 *vs.* 29/100, $p = 0.047$). Histological analysis of FGF2-treated wounds showed a significant increase in the number of fibroblasts and capillaries.

### Diabetic foot

Diabetic foot is a serious complication of diabetes and an important cause of diabetes-related disability. When diabetic foot develops, the patient's feet are prone to injury, infection, ulcers and gangrene.

*Uchi et al. (2009)* conducted a randomized, double blind, dose-ranging, placebo-controlled trial to examine the clinical efficacy of FGF2 in the treatment of diabetic ulcers. Patients' diabetic ulcers were randomized into a placebo group (n = 51), a 0.001% (w/v) FGF2 treatment group (n = 49) and a 0.01% (w/v) FGF2 treatment group (n = 50), with the primary outcome being the percentage of patients showing a 75% or greater reduction in the area of ulcer. The area of ulcer decreased by 75% or more in 57.5% (27/47), 72.3% (34/47), and 82.2% (37/45) in the placebo, 0.001% (w/v) FGF2 and 0.01% (w/v) FGF2 groups, with significant differences between the 0.01% (w/v) FGF2 treatment and placebo groups ($p = 0.025$). Cure rates were 46.8%, 57.4%, and 66.7% in the placebo, 0.001% (w/v) FGF and 0.01% (w/v) FGF2 groups. This trial showed that FGF2 accelerates healing of diabetic ulcers.

### Critical limb ischaemia

*Kumagai et al. (2015)* conducted a phase I-IIa trial, investigating the use of a sustained release system of FGF2 using a biodegradable gelatin hydrogel in patients with critical limb ischaemia. The measured transcutaneous pressure, distance walked in 6 minutes, rest pain scale and cyanotic pain scale showed significant improvement at 24 weeks post-treatment as compared to pre-treatment.

### Other applications

Repair of cerebrospinal fluid leakage is difficult, which is especially so when a large fistula, with concomitant mucosal damage and infection, has developed from repeated transsphenoidal operations. *Kubo et al. (2005)* reported a 27-year-old woman with intractable cerebral spinal fluid rhinorrhea who had undergone repeated operations for a relapsing Rathke's cleft cyst. They repaired the sellar floor defect using mucosal flaps via an endonasal endoscopic approach and occluded the fistula by applying FGF2 to the area to promote granulation. FGF2 was repeatedly applied endoscopically to the mucosal flaps, which turned into granulation-like tissue, and complete mucosal covering was attained. This method of treating the intractable fistula with mucosal flaps and FGF2 may present a new clinical application of FGF2 and should be examined in a large number of patients in the future.

The mucosa of the vocal folds atrophies with age causing glottal insufficiency, which is difficult to treat. *Hirano et al. (2008)* reported a case of a patient, with atrophied vocal folds, who was treated with FGF2 injections into the folds under local anaesthesia.

The atrophy of the vocal fold improved within a week following the injection and the glottic gap disappeared. Aerodynamic and acoustic parameters also showed remarkable improvement, when measured. Subsequently, a trial (*Hirano et al., 2012*) demonstrated that this therapy might be safe and effective in the treatment of age-related vocal fold atrophy.

## PROSPECTS

In Europe and N America, the substantial investment by cancer sources into growth factors such as the FGFs has resulted in oncology directed clinical translation, in the form of FGFR inhibitors (*Carter, Fearon & Grose, 2015*; *Turner & Grose, 2010*). In contrast, the biotechnology drive in China resulted in exploitation of engineered FGF ligands to repair and regenerate damaged tissue in a wide range of settings, with Japan having clinical experience in both areas. We have not been able to identify reports of adverse reactions to treatment with FGF ligands–these undoubtedly occur, but the frequency or their severity may be too low and confounded by the underlying medical condition, such that they have not appeared in the case literature. In any event, it is clear that the Chinese and Japanese experience with FGF ligands as biologics in repair and regeneration clinical scenarios has been an outstanding success; FGFs in China have progressed from engineered biotechnology products (*Wu et al., 2005*; *Wu et al., 2004*; *Yao et al., 2006*; *Zhao et al., 2004*) to the Chinese (*Pharmacopeia, 2015*). Many of the conditions, e.g., diabetic foot, make important and growing demands on healthcare systems and carry considerable socioeconomic costs. Thus, Western medical practice may usefully follow where China and Japan have led and explore the use of FGF ligands as repair and regeneration agents. The realisation of the clinical potential of the FGF communication system outside of oncology has been long overdue in the West. However, Western industry is now actively engaged in the development of FGF therapeutics. This includes development of FGF21 based therapeutics for metabolic syndrome (*Kharitonenkov & Shanafelt, 2008*; *Kharitonenkov & Shanafelt, 2009*; *Zhang & Li, 2014*), of FGF7 for oral mucositis (*Goldberg et al., 2013*) and of FGF18 in osteoarthritis (*Carli et al., 2012*; *Mori et al., 2014*), it is likely that we will see FGF biologics in clinical use in the West, as well as in the East. Indeed, the use of FGF18 to treat osteoarthritis is progressing through clinical trials (*Lohmander et al., 2014*).

## ADDITIONAL INFORMATION AND DECLARATION

### Funding

DGF, CS and YL received funding from the Cancer and Polio Research Fund and North West Cancer Research. QMN received a National Institute for Health Research Lectureship in General Surgery. DGF is Co-I on the National Institute for Health Research Liverpool Pancreas Biomedical Research Unit. TKK received a NC3R PhD studentship. The funders had no role in study design, data collection and analysis, decision to publish, or preparation of the manuscript.

## Competing Interests

The authors declare that they have no competing interests.

## Author Contributions

- Quentin M. Nunes wrote the paper, reviewed drafts of the paper, reviewed Chinese clinical papers.
- Yong Li wrote the paper, prepared figures and/or tables, reviewed drafts of the paper, translated Chinese papers.
- Changye Sun wrote the paper, prepared figures and/or tables, reviewed drafts of the paper, translated Chinese papers.
- Tarja K. Kinnunen wrote the paper, reviewed drafts of the paper.
- David G. Fernig wrote the paper, reviewed drafts of the paper.

## Data Deposition

The research in the article did not generate any raw data.

## Supplemental Information

Supplemental information for this article can be found online at http://dx.doi.org/10.7717/peerj.1535#supplemental-information.

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
