# Peer review of "Fibroblast growth factors as tissue repair and regeneration therapeutics"

_PeerJ, doi:10.7717/peerj.1535_

## Round 0.1 · original submission · Minor Revisions

Dear Authors,

Your manuscript was reviewed by three independent expert reviewer. While all three reviewers have appreciated the manuscript, there are some comments which need to be taken care of. I believe that these comments will be useful to improve the manuscript further.

With best regards

·

Basic reporting

The review by Numes et al on FGFs as tissue repair and regeneration therapeutics is superbly written. One major criticism is that the review focusses mostly on the clinical use of FGF2 and not other FGFs (FGF21 and FGF18 are quickly mentioned in the conclusion, nothing is mentioned for FGF7 and FGF10)... So, i would suggest to either change the title or expand on the other relevant FGFs.
In addition, the part on the FGFs in C. elegans, could have been removed (as it appears a little bit odd in the review).

Few typos in the text
Line 187: large number Of signalling pathways
Line 332 (REFs Preta): reference needs formatting.

Experimental design

N/A

Validity of the findings

N/A

Additional comments

Great review David. It would be nice to maybe have a table with all the references of our chinese colleagues and their main finding to be able to visualise the journals where these clinical reports are published (with maybe some validation studies done in the West to support these results). One main aspect as you pointed out is the lack of accessibility of the chinese literature.

·

Basic reporting

The article feels very much like it has been written by different people - the initial scene setting is beautifully written and very interesting, then there seems to be a rather large section on C. elegans, which fits with author expertise but sits rather oddly without any other animal models being considered. The last therapeutic section seems to have been collated from a series of paper abstracts with very little substance - this is the section that needs work - very often there is information on concentration/dose/controls missing and it feels like a list rather than having any critique. This is potentially a very important part of the review, so it makes sense to put more detail into it - would also be well supported by inclusion of a summary table of studies.

Experimental design

As above - although a review, it would be good to include more experimental detail from the papers being summarised.

Validity of the findings

Good subject to points above

Additional comments

It reads as rather an eclectic mix of studies, but there is lots of interesting information contained within. I think that, with addressing typos/comments attached in PDF, it will make a significant impact as a review article - no doubt that there has been a failure to communicate the breadth of research into FGF therapeutics in China, and this will help.

Reviewer 3 ·

Basic reporting

1. Lines 4-18. Suggest authors should be citied the references.
2. Lines 127. Itoh & Ornitz 2011b, the reference was marked correct?
3. Lines 119-129. the paragraph similar to the article (Fibroblast growth factors: from molecular evolution to roles in development, metabolism and disease).

Experimental design

No Comments

Validity of the findings

No Comments

Additional comments

There are some similarities between the review and the article (Fibroblast growth factors: from molecular evolution to roles in development, metabolism and disease)

---

## Round 0.2 · accepted · Accept

Dear Authors,

I am happy to inform you that both the reviewers have expressed their satisfaction on the revised version. Based on their comments, your manuscript has been accepted for publication in PeerJ.

·

Basic reporting

Excellent

Experimental design

Excellent

Validity of the findings

Excellent

Additional comments

The authors have made significant improvements and the manuscript is ready to go - the new table is an excellent addition and is sure to encourage a lot of interest, as well as citations. Great review.

Reviewer 3 ·

Basic reporting

No Comments

Experimental design

No Comments

Validity of the findings

No Comments

Additional comments

No Comments